# High Poisonous Cd Ions Removal by Ru-ZnO-g-C$_3$N$_4$ Nanocomposite: Description and Adsorption Mechanism

Mukhtar Ismail [1,*], Abuzar Albadri [2], Mohamed Ali Ben Aissa [1], Abueliz Modwi [1,*] and Sayed M. Saleh [2,3]

1 Department of Chemistry, College of Science and Arts, Qassim University, Ar-Rass 51921, Saudi Arabia
2 Department of Chemistry, College of Science, Qassim University, Buraidah 51452, Saudi Arabia
3 Chemistry Branch, Department of Science and Mathematics, Faculty of Petroleum and Mining Engineering, Suez University, Suez 43721, Egypt
* Correspondence: muk.mohamed@qu.edu.sa (M.I.); ab.khalid@qu.edu.sa (A.M.)

**Abstract:** Ru-ZnO-g-C$_3$N$_4$ nanocomposite was made using a straightforward ultrasonication method and evaluated for its potential to remove Cd ions from aqueous environments. X-ray diffraction analysis confirms composite production with an average crystalline size of 6.61 nm, while transmission electron microscopy results indicate nanosheet-like nanomaterials with uniform elements distribution. Measurements of N2 adsorption–desorption reveal the creation of a mesoporous structure with a BET surface area of approximately 257 m$^2$/g. Fourier converted infrared reveals vibrational modes for O-H, amino groups, triazine, and Ru-ZnO. In contrast, X-ray photoelectron spectroscopy investigation reveals the presence of the elements Ru, Zn, O, N, and C. Ru-ZnO-g-C$_3$N$_4$ nanocomposite has remarkable adsorption efficiency for aqueous Cd ions, achieving 475.5 mg/g in 18 min. This study reveals that the Ru-ZnO-g-C$_3$N$_4$ nanocomposite may be used as an effective and reusable adsorbent for removing Cd ions during wastewater treatment and, possibly, for eliminating other toxic metal ions.

**Keywords:** Ru-ZnO-g-C$_3$N$_4$ nanocomposite; poisonous Cd ions; removal kinetic; adsorption mechanism





## 1. Introduction

Emissions of inorganic pollutants into the environment are a major cause for concern due primarily to their transformation into more harmful compounds [1]. Cadmium is a metal with toxic characteristics; long-term, superficial intake of cadmium has adverse impacts on the health of people, including the development of diabetes, hypertension, and cancer [2]. Diet is the primary source of cadmium exposure, second only to smoking; multiple studies have linked prolonged dietary consumption of cadmium with a higher risk of renal dysfunction and osteoporosis [3]. Dietary exposure to cadmium occurs when crops that are consumed as food absorb cadmium from agricultural soil. The high persistence of cadmium in soil and the high soil-to-plant transfer rates facilitate this process [4]. The presence of cadmium in natural groundwater is a global issue that has garnered considerable attention [1,2]. Multiple industrial and agricultural processes and mining activities have raised the quantity of harmful heavy metals in various environmental components, such as water, wastewater, and soils, around the globe. Hazardous metals in water negatively affect ecosystem function and may pose concerns to human health [3–6].

Several methods, including biological treatments, filtration, and adsorption [6–9], have been developed to remove Cd ions from wastewater. However, very few irrigation water treatment options are simple, inexpensive, and environmentally beneficial. Adsorption has been studied as an option that can meet these requirements, and efforts have been made over the past decade [10,11] to develop acceptable materials for the adsorptive removal of Cd ions [12,13]. However, the development of such materials is hampered by the fact that the two contaminants have separate physiochemical properties in agricultural waterways. In addition, cadmium typically exists as a divalent cation, i.e., Cd ions. In treating inorganic

contaminants in the aquatic system [14,15], nanomaterials are particularly appealing and are being investigated on a large scale. Specifically, the features of nanocomposites, such as $RuO_2$-ZnO, $Y_2O_3$-ZnO, $CaMgO_2$@g-$C_3N_4$, and $Fe_3O_4$@$SiO_2$-$NH_2$ are widely employed in water purification due to their new physicochemical characteristics, which can also be modified by doping with various materials to meet special needs and purposes [16–19].

This research intends to generate a high surface nanocomposite by a simple method to solve and remove the pollution problem of Cd ions in aquatic systems. A ternary Ru-ZnO-g-$C_3N_4$ nanocomposite for removing Cd ions from an aqueous solution is prepared and evaluated. The efficacy of removal factors for adsorption capacity, including starting Cd ions concentration, pH, and contact time, have been examined. In addition, the isotherms of adsorption, kinetic investigations, and recyclability have been discussed. A feasible mechanism for eliminating Cd ions from the Ru-ZnO-g-$C_3N_4$ nanocomposite surface is shown through FTIR analysis.

## 2. Results and Discussions

### 2.1. Ru-ZnO-g-$C_3N_4$ Nanosorbent Characteristics

As shown in Figure 1, the crystalline phase of the Ru-ZnO@g-$C_3N_4$ nanocomposite was evaluated using X-ray diffraction (XRD) techniques. The peaks of g-$C_3N_4$ were located at 12.64° and 27.22°, revealing that the distance between the interlayer structural module and the separation of interconnected aromatic systems related to planes (100) and (002), respectively. The Ru-ZnO nanocomposite exhibited peaks at 2θ = 31.98, 34.64, 36.45, 47.75, 56.84, 63.12, 66.30, 68.15, and 69.28° that might be attributed to the (100), (002), (101), (102), (110), (103), (200), (112), and (201) surfaces of the wurtzite hexagonal ZnO structure, respectively, conferring to JCPDS card (No.36-1451) [20]. In particular, additional peaks corresponding to 2θ = 27.9, 34.92, and 54.35° are identified, delineating the $RuO_2$ rutile phase's (110), (101), and (211) planes in accordance with JCPDS card No. 88-0308) [5]. The XRD pattern indisputably demonstrates that the composite is composed of g-$C_3N_4$ and Ru-ZnO diffraction peaks. This research demonstrated that Ru-ZnO nanoparticles were deposited on g-$C_3N_4$ in the nanocomposite, reducing the d-space of conjugated aromatic systems with a crystalline size of 6.61 nm.

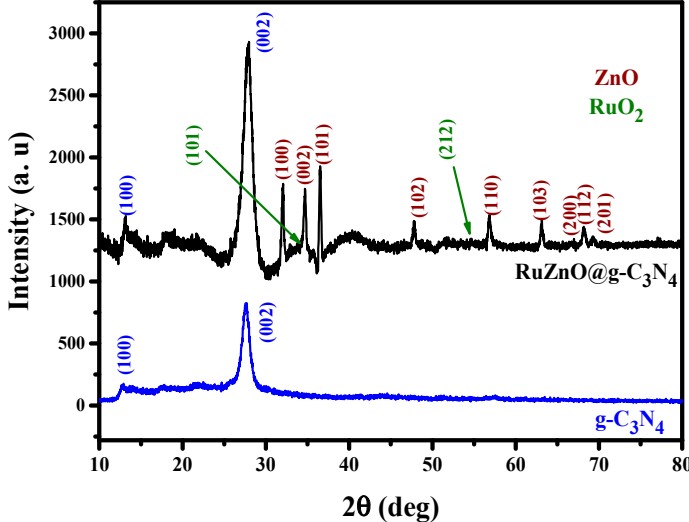

**Figure 1.** XRD patter of g-$C_3N_4$, and Ru-ZnO@g-$C_3N_4$.

The N–H bond stretching vibration, in conjunction with the OH stretching of the adsorbed moisture, is responsible for the broad absorption band that may be found in the range of 3300–3050 $cm^{-1}$ [21,22]. It is possible to attribute the normal stretching of C–N and C–N bonds to a group of peaks that fall within the range of 1631 to 1232 $cm^{-1}$ [23]. The band with a center frequency of 808 $cm^{-1}$ is a characteristic feature of the bending

vibration of the s-triazine ring. This band indicates the presence of hexazine units in the structure of the as-prepared Ru-ZnO-g-C$_3$N$_4$ nanocomposite [24]. In addition, the band at 731 cm$^{-1}$ can be assigned to the stretching vibrations of ZnO [25]. The bonding pattern of the g-C$_3$N$_4$ bonding system has been preserved in the FTIR spectrum of the composite (Figure 2), and it can be seen alongside the Ru-Zn-O peak at 490 cm$^{-1}$, which indicates that the Ru-ZnO is bonding to the g-C$_3$N$_4$.

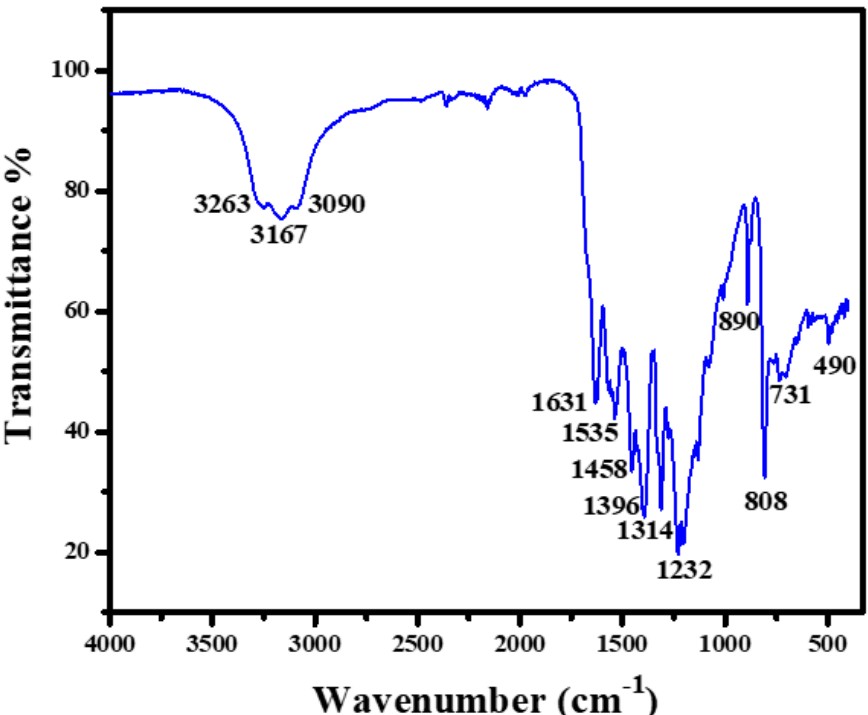

**Figure 2.** FTIR of Ru-ZnO-g-C$_3$N$_4$ nanosorbent.

A substantial number of adsorption sites must be created to ensure a suitable adsorbent, such as Ru-ZnO-g-C$_3$N$_4$ nanocomposite. In other words, the material's surface area, pore volume, and size should be sufficient. The nitrogen adsorption–desorption analysis (Figure 3 inset, pore size) demonstrates that the Ru-ZnO-g-C$_3$N$_4$ nanocomposite is a mesoporous material with an IUPAC type IV adsorption isotherm. The isotherm is associated with a type H1 hysteresis, indicating a narrow distribution of uniform mesoporous and limited network effects [26,27]. The transmission electron micrograph (Figure 4) reveals the presence of pores and the appearance of a large aggregation produced by the connectivity of Ru-ZnO nanoparticles with g-C$_3$N$_4$ nanosheets. The Ru-ZnO-g-C$_3$N$_4$ nanocomposite has a surface area, total porous volume, and average pore radius of 257 m$^2$/g, 0.499 cc/g, and 15.778 Å, respectively. It is anticipated that the adsorbent's huge surface area and porosity will expose many adsorbent surfaces, leading to a high adsorption efficiency.

As demonstrated in Figure 4a, the TEM image of the produced Ru-ZnO-g-C$_3$N$_4$ nanocomposite reveals distinctive nanosheet-like two-dimensional nanostructures with a curved thickness of approximately 25 nm. As can be seen, the average particle size of the incorporated Ru-ZnO nanostructures in the Ru-ZnO-g-C$_3$N$_4$ nanocomposite is between 10 and 30 nm. The EDX spectrum (Figure 4b) reveals typical peaks of Ru, Zn, N, O, and C, verifying the purity of the produced composite. The matching chemical composition of the produced Ru-ZnO-g-C$_3$N$_4$ nanocomposite is shown in the inset Table of Figure 4b. As shown in Figure 4c–f,h,i, the elemental scanning analysis for C, N, Ru, Zn, and O in the as-fabricated Ru-ZnO-g-C$_3$N$_4$ nanocomposite agglomeration reveals a generally homogeneous dispersion. On the elemental maps, a brighter zone implies a higher elemental ratio. This result suggests that the Ru-ZnO-g-C$_3$N$_4$ nanocomposite has produced a homogeneous dispersion.

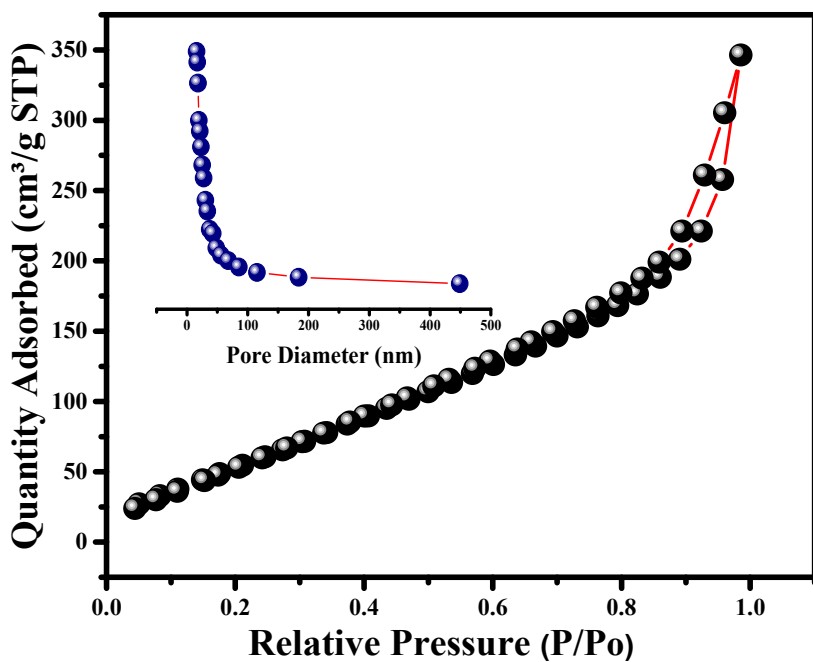

**Figure 3.** BET surface area (inset pore distribution).

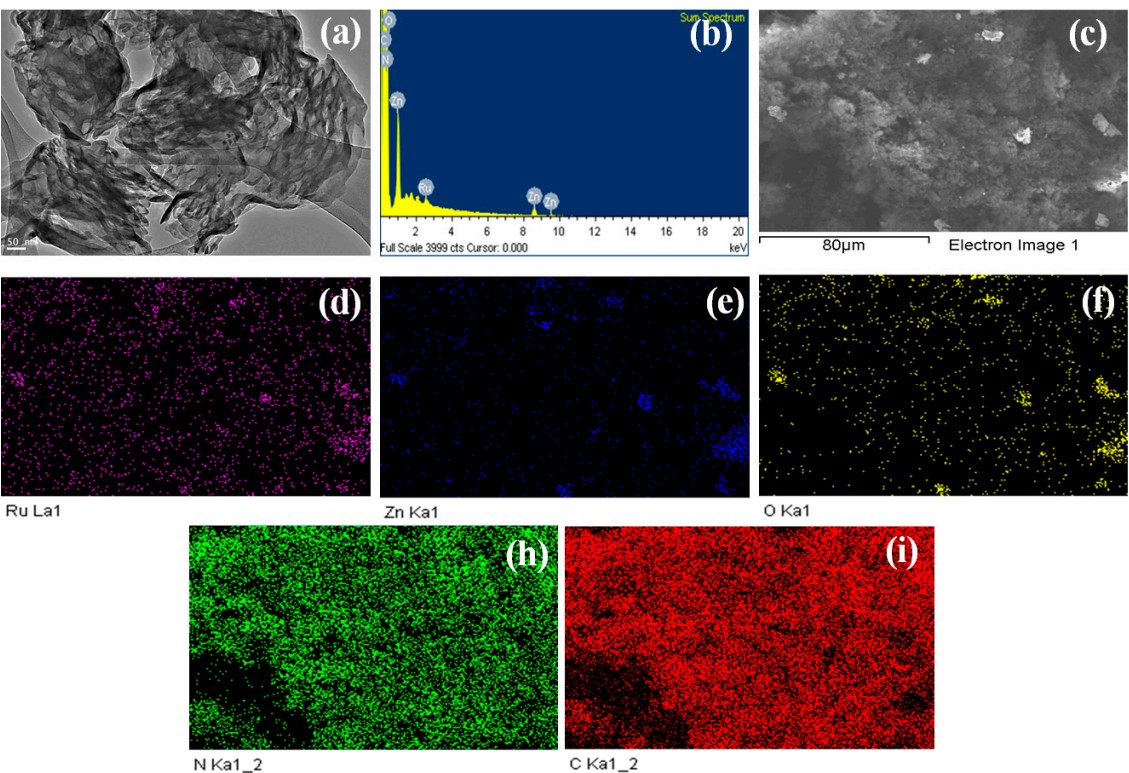

**Figure 4.** (**a**) TEM image, (**b**) EDX and (**c**–**f**,**h**,**i**) elemental mapping of nanocomposite.

The chemical oxidation states of C, N, O, Zn and Ru in the Ru-ZnO-g-$C_3N_4$ nanocomposite were investigated by the XPS technique. As given in the N 1s spectra (Figure 5a), the peak at 396.4 eV was assigned to sp2-hybridized nitrogen (C–N–C). The peaks at 284.4 eV and 285.8 eV in the C 1s spectrum (Figure 5b) correspond to the N–C–N coordination [28]. Figure 5c depicts the XPS spectra of Zn 2p core levels, revealing two symmetric peaks at 1019.0 and 1047.4 eV. These peaks are ascribed to Zn 2p3/2 and Zn 2p1/2, respectively,

and indicate the $Zn^{2+}$ oxidation state [29]. Figure 5d depicts the O1s spectra on the surface of Ru-ZnO-g-$C_3N_4$ nanocomposite. The O1s signal was deconvoluted into three peaks at 530.1, 528.3, and 526.6 eV, corresponding to lattice oxygen in ZnO, $RuO_3$ ($Ru^{6+}$), and $RuO_2$ ($Ru^{4+}$), respectively [30]. The XPS spectra of Ru 3d core levels exhibited a peak at 285.2 eV, assigned to Ru 3d5/2 (Figure 5e). After deconvoluting, two small peaks were specified at 284.8 eV and 285.7 eV, indicating the two oxidation states $Ru^{4+}$ and $Ru^{6+}$, respectively [31].

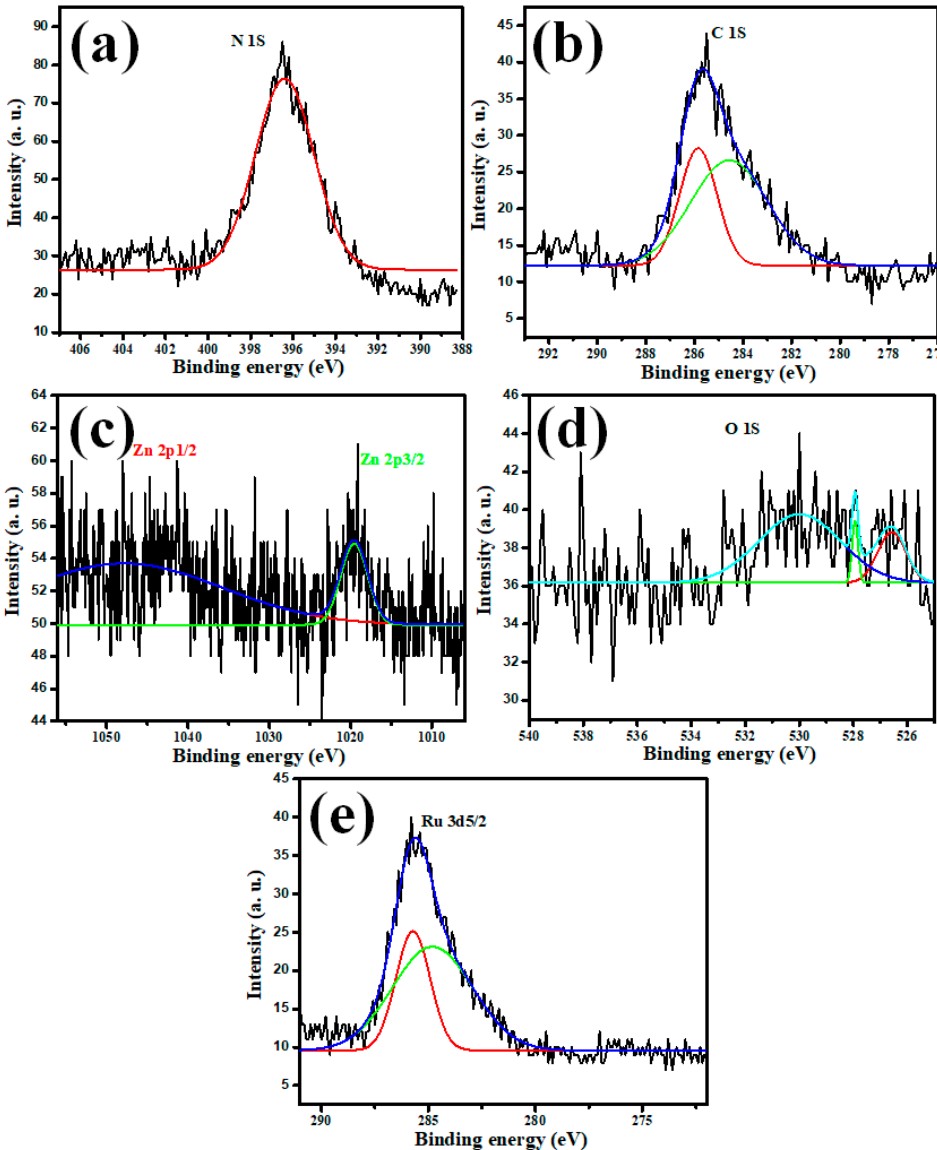

**Figure 5.** XPS analysis of (**a**) N 1s, (**b**) C 1s, (**c**) Zn 2p, (**d**) O1s, and (**e**) Ru 3d elemental for Ru-ZnO-g-$C_3N_4$ nanocomposite.

### 2.2. Adsorption Capability of Ru-ZnO-g-$C_3N_4$ Nanocomposite

#### 2.2.1. Impact of Initial Cd (II) Concentration

When the influence of the initial concentration of Cd ions was examined in the range of 5 to 200 ppm under optimized solutions, Ru-ZnO-g-$C_3N_4$ nanocomposite dose (10 mg), pH 7, fixed volume of Cd ions solution (25 mL), room temperature, and a 24-h contact time were achieved. The proportion and optimal adsorption capacity of Cd ions on Ru-ZnO-g-$C_3N_4$ nanocomposites are depicted in Figure 6. As the Cd ions concentration rises, the adsorbate quantity climbs steadily from 11.87 mg/g to 370.86 mg/g, while the removal efficiency remains extremely high at 97.88%. In this instance, the fundamental driving force that increases the initial Cd ion concentration overcomes any barrier to

Cd ion migration from the solution. The obtained fractional adsorption turns out to be concentration dependent.

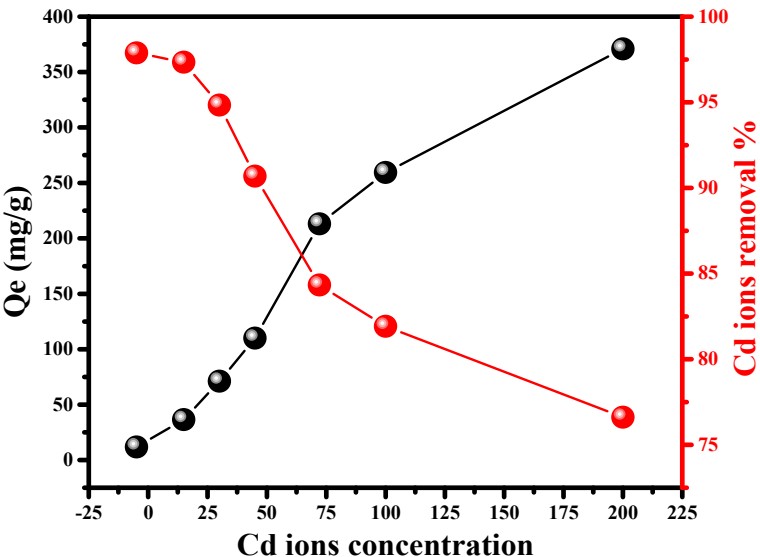

**Figure 6.** Influence of Cd (II) initial concentration.

### 2.2.2. Impact of Difference pH on Cd (II) Removal

The pH value is considered to be significant when comprehending the intensity of surface reactions between adsorbate (Cd ions) and Ru-ZnO-g-C$_3$N$_4$ nanocomposites. The influence of pH on Cd ions adsorption efficiency was studied between the pH range of 1.0 and 8.0, with the optimal adsorption capacity achieved at pH 5, as shown in Figure 7. Below pH 5, the adsorbent surface is significantly protonated, leading to poor adsorbate–adsorbent interactions. Similar results were found in trials with similar designs [32,33]. The solubility of metal ions, such as Cd ions, depends on pH. Cd ions are very soluble as Cd ions free ions and Cd (OH)+ at lower pH. Moreover, Cd ions precipitate as metal hydroxide Cd (OH)$_2$ at pH values greater than 7.5 [2,34]. These results demonstrate that the quantitative removal capacity increases with the pH value of the solution until it reaches 5.

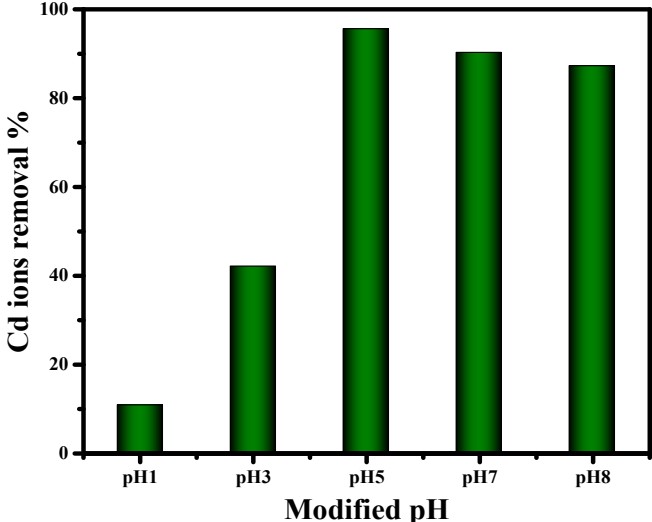

**Figure 7.** Effect of disparity pH on Cd (II) adsorption.

### 2.2.3. Adsorption Isotherms Modeling

To determine the maximal adsorption capacity, adsorption isotherm tests were conducted at pH = 7 with different initial concentrations of Cd ions. As depicted in Figure 8a,b, the amount of Cd ions adsorbed on the Ru-ZnO-g-$C_3N_4$ nanocomposite increases as starting Cd ion concentration varies. The Freundlich and Langmuir adsorption models were used to replicate the experimental data on the adsorption of Cd ions onto the Ru-ZnO-g-$C_3N_4$ nanocomposite. The Langmuir model (Figure 8a) is better suited than the Freundlich model for modeling the adsorption of Cd ions. In addition, Table 1 contains the equations and parameters of the Langmuir and Freundlich adsorption models, where the maximal adsorption capacity of Cd ions at pH 7 and room temperature is 370 mg/g. The correlation coefficients ($R^2$) of the Langmuir model for Cd ions are approximately 0.9958, which is greater than the Freundlich model (0.9911) and consistent with the simulations presented in Figure 8b.

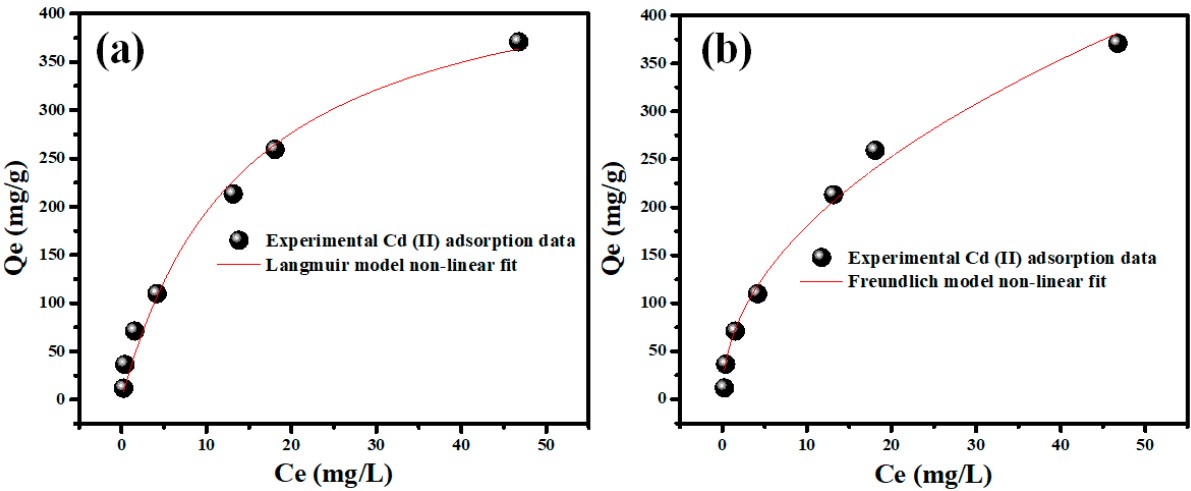

**Figure 8.** Cd (II) adsorption fitted with nonlinear Langmuir (**a**) and Freundlich model (**b**).

**Table 1.** Cd ions equilibrium adsorption and kinetics models parameters computed.

| Adsorption Model | Langmuir | Freundlich | PFO | PSO |
|---|---|---|---|---|
| Parameter | $Q_{max}$ = 475.5 mg/g | n = 59.07 | Q = 98.34 | $Q_{Cal}$ = 103.7, $Q_{Exp}$ = 102.5 |
| | $K_L$ = 0.069 | $K_f$ = 0.485 | $K_1$ = 0.109 | $K_2$ = 0.0064 |
| $R^2$ | 0.9958 | 0.9911 | 0.9489 | 0.9945 |
| Chi-Sqr | 295.85 | 186.92 | 21.91 | 19.034 |

### 2.2.4. Contact Time and Adsorption Kinetics Modeling

To acquire information about the mechanism governing the sorption of Cd ions, kinetic models utilizing many model equations are generally conducted (Figure 9a–d). Figure 9a depicts the influence of contact time on the removal of Cd ions at ambient temperature. With stirring periods ranging from 5 to 1440 min and an initial metal ion concentration of 60 mg, the adsorption of Cd ions onto the nanocomposites was explored. In less than 18 min, the removal of Cd ions as a function of contact time reaches equilibrium. Due to the high number of active sites on the Ru-ZnO-g-$C_3N_4$ nanocomposite surface, the initial sorption process is extremely rapid, reaching $h_0$ = 2.05 mg g$^{-1}$ min$^{-1}$. After the equilibrium phase, the concentration of the active site gradually drops, and the percentage removal reaches equilibrium substantially more slowly. Accordingly, 18 min can be regarded as a short time to attain equilibrium.

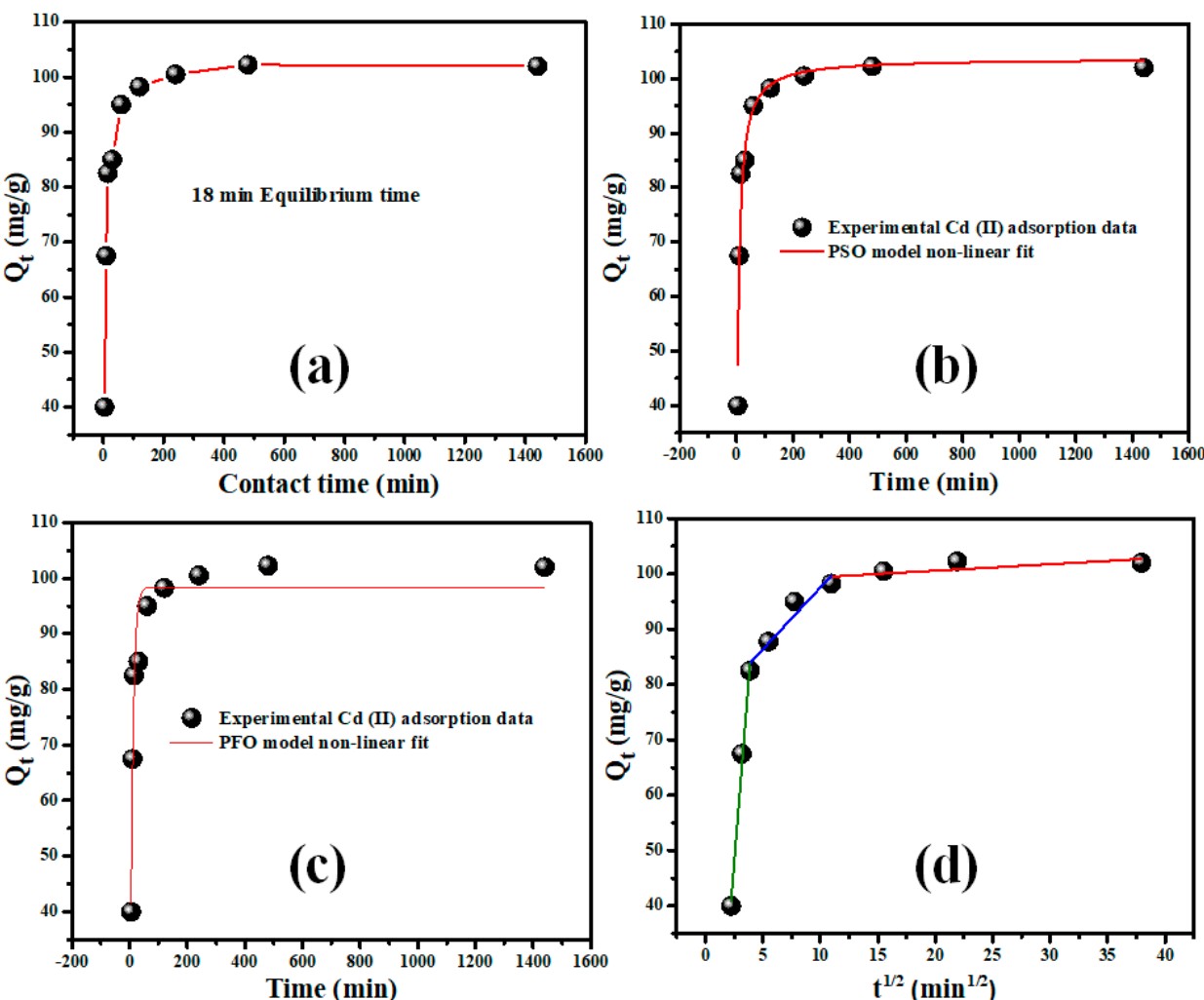

**Figure 9.** (**a**) Contact time, (**b**) PFO (**c**) PSO kinetics and (**d**) IPD models for Cd (II) adsorption (**c**).

To study the Cd ions adsorption rate, two well-known kinetic models were used to comprehend the kinetics of Cd ion adsorption on Ru-ZnO-g-C$_3$N$_4$ nanocomposite. The non-linearized versions of the pseudo-first-order [35] (Equation (1)) and pseudo-second-order [36] (Equation (2)) kinetic models are as follows:

$$q_t = q_e \left(1 - e^{-1k_1 t}\right) \tag{1}$$

$$q_t = \frac{t k_2 q_e^2}{k_2 q_e t + 1} \tag{2}$$

where $q_t$ and $q_e$ (mg g$^{-1}$) represent the uptake capacity of the adsorbent at time $t$ and equilibrium, accordingly; $k_1$ (min$^{-1}$) and $k_2$ (g mg$^{-1}$ min$^{-1}$) represent the rate constants in the pseudo-first-order rate formula and pseudo-second-order rate formula, respectively.

Figure 9b,c depicts the pseudo-first-order and pseudo-second-order kinetic graphs for Cd ions adsorption. Table 1 displays the estimated and experimental kinetic parameters. Because when adsorption data are fitted into the pseudo-second-order kinetic model, the $R^2$ values obtained for Cd ions adsorption are greater than 0.99. The estimated $q_{max}$ and the experimental $q_{max}$ are comparable. These results suggest that the pseudo-second-order kinetic model more accurately characterizes the kinetics of Cd ion adsorption on Ru-ZnO-g-C$_3$N$_4$ nanocomposite surfaces.

If sufficient information is provided, the pseudo-second-order kinetic model can be employed to extrapolate all stages of the sorption process, such as outer film diffusion, sorption, and internal particle diffusion. However, this model cannot explain the specific adsorption mechanism [36]. Consequently, the collected data were evaluated with the intraparticle diffusion kinetic model (Figure 9d). Numerous studies indicated that the intraparticle diffusion graph might display multi-linearity, indicating that two or more steps may occur during the adsorption process [37]. The results reveal that three straight lines represent the majority of data points and that the plots do not intersect the origin.

As qt varies linearly with $t_{1/2}$ upon removal of Cd ions from the Ru-ZnO-g-C$_3$N$_4$ nanocomposite surface, the intra-particle transport kinetic model is validated. C identifies the thickness of the border layer. The substantial value of the constant in Table 2 suggests that the solution boundary layer has a significant impact on adsorption [38–40]. According to $k_{dif1}$, $> k_{dif2} > k_{dif3}$, the initial stage of Cd ion elimination has a higher rate than the second and third stages (Table 2). The quick rate of the initial stage may be attributable to the transport of ions from the solution to the surface of the outer nanostructures via the boundary layer. Concurrently, the subsequent stage mirrors the final equilibrium stage when intra-particle diffusion begins to decrease due to the solute's low concentration differential and fewer accessible diffusion pores. In addition, the increasing quantity of component C in the second stage indicates the presence of a boundary layer, validating the role of intraparticle diffusion in the Cd ions uptake by Ru-ZnO-g-C$_3$N$_4$ nanocomposite [17,41].

**Table 2.** Cd ions Intra-particle kinetic transport model and derived magnitudes.

| Step | 1 | 2 | 3 |
|---|---|---|---|
| $k_{dif}$ (mg g$^{-1}$ min$^{-1/2}$) | 26.15 | 2.233 | 0.117 |
| C | 17.54 | 25.21 | 98.21 |
| R$^2$ | 0.9923 | 0.9890 | 0.5682 |
| RSS | 7.900 | 10.178 | 4.371 |

### 2.3. Adsorption Mechanism

The Cd ions adsorption mechanism of the Ru-ZnO-g-C$_3$N$_4$ nanocomposite has been clarified using the FTIR spectrum. Figure 10a illustrates that the FT-IR spectra of Ru-ZnO-g-C$_3$N$_4$ and Cd@Ru-ZnO-g-C$_3$N$_4$ obtained between 500 and 4000 cm$^{-1}$. Ru-ZnO-g-C$_3$N$_4$ spectral bands can be identified as follows: the bandwidth between 3000 and 3400 cm$^{-1}$ corresponds to the NH stretching mode of the terminal amino group. The bands at 1228, 1312, and 1409 cm$^{-1}$ relate to the aromatic C–N stretching mode, and those at 1571 and 1652 cm$^{-1}$ belong to the C≡N stretching mode [42]. The peak at 889 cm$^{-1}$ is a triazine ring mode, which is a comparatively common carbon nitride mode [43]. As depicted in Figure 10a, the triazine ring mode and aromatic C–N stretching modes of Ru-ZnO-g-C$_3$N$_4$ have altered positions after the adsorption of Cd ions [44,45]. This result suggested that functional groups of Ru-ZnO-g-C$_3$N$_4$ (N-H and CN) and delocalized electron systems of the triazine ring (C$_3$N$_3$) were responsible for the elimination of Cd ions. Figure 10b depicts a possible pathway for the adsorption of Cd metal ions into the Ru-ZnO-g-C$_3$N$_4$ nanocomposite.

### 2.4. Assessment Study

To demonstrate the extraordinary Cd ion adsorption capability of Ru-ZnO-g-C$_3$N$_4$ nanocomposite, Table 3 compares the results obtained with those of other adsorbent materials that have been previously reported. Under optimal conditions, it is evident that the produced Ru-ZnO-g-C$_3$N$_4$ nanocomposite has outstanding efficacy in removing Cd ions, with an adsorptive capacity of 475.5 mg g$^{-1}$ reached in just 18 min. This result is mostly owing to the mesoporous characteristic, nanostructure, and significant surface area of 257 m$^2$ g$^{-1}$ of the produced material. This cost-effective nanocomposite has the

potential to eliminate other hazardous metals and organic pollutants. In line with the pseudo-second-order model, the rate-determining stage is regarded as chemical adsorption that involves the adsorbent/adsorbate of electrons between the adsorbent and adsorbate. The high regression coefficient ($R^2$ = 0.9958) of the Elovich model provides corroborating evidence for the chemisorption character of the Cd ions' adsorption by Ru-ZnO-g-C$_3$N$_4$. The nanostructures of the synthesized Ru-ZnO-g-C$_3$N$_4$ can be used as a suitable adsorbent for aqueous cadmium (II) ions due to their simplicity of manufacture, high adsorption efficacy, recovery ability, and reusability.

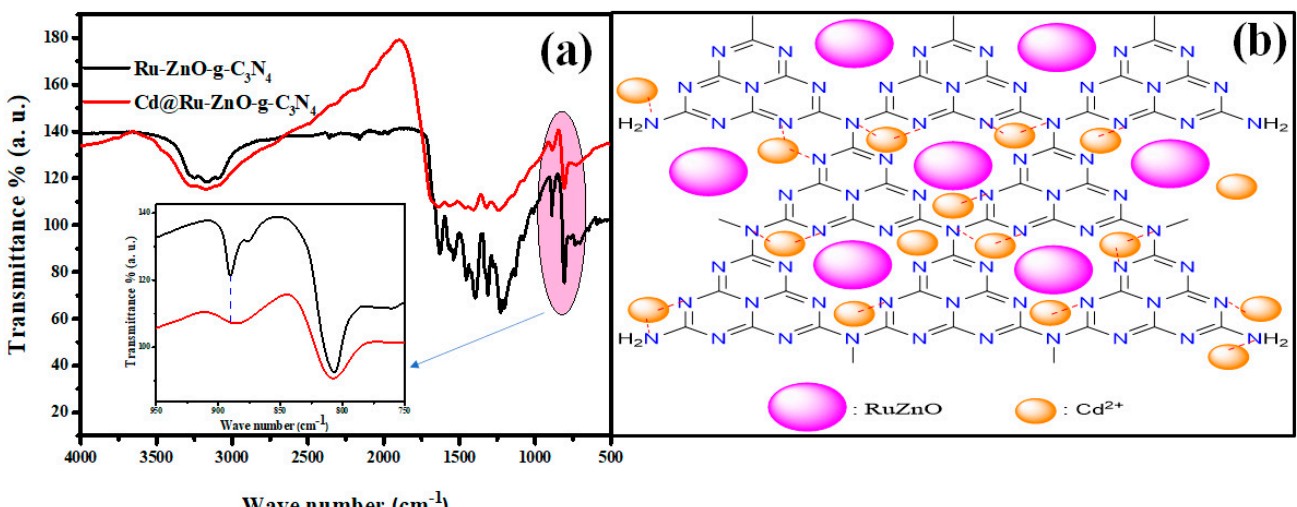

**Figure 10.** FTIR spectra of (**a**) before and after Cd ions into Ru-ZnO-g-C$_3$N$_4$ and (**b**) suggested removal mechanism.

**Table 3.** Monitoring characteristics for the adsorption of Cd ions onto the Ru-ZnO-g-C$_3$N$_4$ nanocomposites in comparison to other adsorbents and nanostructures.

| Materials Used | Pseudo-Second-Order | | Langmuir Isotherm | | Optimal pH | S$_{BET}$ (m$^2$/g) | D (nm) | Ref. |
|---|---|---|---|---|---|---|---|---|
| | k$_2$ (g/mg/min) | R$^2$ | q$_m$ (mg/g) | R$^2$ | | | | |
| Zeolite X | 0.002 | 0.999 | 62.814 | 0.953 | 6.5 | - | 94.85 | [46] |
| Geopolymers/Chitosan | 0.00012 | 0.982 | 166.11 | 0.992 | 8 | - | - | [47] |
| Binary Mg–Si hybrid oxide | 0.033 | 0.999 | 18.790 | 0.998 | 7 | 540 | 56.4 | [48] |
| Chitosan/CaCO$_3$ nanoparticles | - | - | 29.41 | 0.980 | 6.5 | - | 60 | [49] |
| Magnetic cellulose nanocomposites | 0.811 | 1.000 | 103.1 | 0.820 | 6 | 7.72 | 30 | [50] |
| White pottery clay | 0.0124 | 0.999 | 26.991 | 0.999 | 5.5 | 56.58 | - | [51] |
| **Ru-ZnO-g-C$_3$N$_4$** | **0.0064** | **0.9945** | **475.5** | **0.9958** | **5.00** | **257** | **6.61** | **This paper** |

## 3. Experimental Methods

### 3.1. Chemicals and Materials

Carbonyl diamide (CH$_4$N$_2$O, ≥99.0%), zinc nitrate hexahydrate (Zn(NO$_3$)$_2$, 6H$_2$O; ≥99.0%), ruthenium (III) chloride (RuCl$_3$; ≥98.0%), sodium hydroxide (NaOH, ≥99%), cadmium nitrate tetrahydrate (Cd (NO$_3$)$_2$. Both 4H$_2$O, ≥98%) and hydrochloric acid (HCl, 37%), purchased from Merck Company, Rahway, NJ, USA, and were used without further purification.

### 3.2. Ru-ZnO-g-C₃N₄ Nanocomposites Construction

The pure g-C₃N₄ reported in our earlier article [52] was manufactured by employing a well-known method. In a muffle furnace, 4.5 g of carbonyl diamide was inserted in a crucible with an insulation material lid and sintered for 120 min at 550 °C at a rate of 10 °C/min in ambient pressure air. The yellow powders were obtained during a period of consistent cooling. For Ru-ZnO nanomaterials, in a 1000 mL beaker at 523 K for three hours, 0.003 moles of zinc nitrate dihydrate solution and 640 mg of RuCl₃ were treated with the saturation solution of pectinase solution. The resulting brown-white foam was then chilled at room temperature for 20 h. Ru-ZnO nonmaterial was produced by drying and annealing the brown-white powder at 320 K for two hours to create Ru-ZnO nonmaterial.

Ru-ZnO-g-C₃N₄ nanocomposites were fabricated using a step-by-step ultrasonication technique in methanol. In 120 mL of methanol, 2760 mg of g-C₃N₄ was balanced and sonicated for 15 min. The g-C₃N₄ methanolic solution was combined with 1200 mg of Ru-ZnO nanoparticles, and the mixture was sonicated for an additional hour. The milky gray solution was evaporated at 368 K for three hours. The collected Ru-ZnO-g-C₃N₄ nanocomposites were annealed for one hour at 423 K.

### 3.3. Ru-ZnO-g-C₃N₄ Nanocomposites Characterizations

The X-ray diffraction (XRD) pattern of Ru-ZnO-g-C₃N₄ nanocomposites was documented utilizing a Rigaku D/max-RA powder diffractometer equipped with a Cu-K radiation source ($\lambda = 1.5418$ Å). The nanomaterial's Brunauer–Emmett–Teller (BET) surface area was calculated by recording N2 adsorption/desorption at 196 °C on a Micro-metrics ASAP 2020 analyzer. A Hitachi H-800 transmission electron microscope (TEM) with dispersive electron X-ray (EDX) spectroscopy was employed for morphological observations and elemental chemistry investigation. X-ray photoelectron spectroscopy (XPS) was used to evaluate the chemical surface properties of the as-fabricated nanocomposite utilizing a Perkin Elmer PHI 550 ESCA/SAM equipped with a monochromatized Al-K X-ray source (hm = 1486.6 eV) and a hemispherical electron analyzer. A Nicolet Nexus 880 FTIR spectrometer and the KBr pellet technique showed FTIR spectra of Ru-ZnO-g-C₃N₄ nanocomposites before and after Cd ion removal to understand the probable adsorption mechanism.

### 3.4. Cd Ions Removal Procedures

Cd ion adsorption isotherms on Ru-ZnO-g-C₃N₄ nanocomposites were evaluated employing batch experiments. In 50 mL glassware, with initial Cd ion concentrations varying from 5 to 200 ppm, 10 mg of Ru-ZnO-g-C₃N₄ nanocomposites sorbent was introduced. For 24 h, the combination suspensions were magnetically stirred. After achieving equilibrium with the aqueous phase, the nanopowder was centrifuged, and atomic absorption spectroscopy (AAS) was used to quantify the remaining Cd ion concentrations in the aliquot (Hitachi Z-8100, Japan). Using the following equations, the amount of adsorbed Cd ions at any time $t$ (min) and the consequent equilibrium values of $q_t$ and $q_e$ (in mg/g) were determined by calculating:

$$q_t = \frac{V(C_0 - C_t)}{m}$$

$$q_e = \frac{V(C_0 - C_e)}{m}$$

where $V$ represents the quantity of the solution ($L$), $C_0$, $C_e$, and $C_t$ are the starting concentration, equilibrium concentration, and concentration, respectively, at any period interval of Cd ions in solution (mg/L), and m is the weight of the Ru-ZnO-g-C₃N₄ nanocomposites (g).

## 4. Conclusions

The present investigation demonstrates that the Ru-ZnO-g-C₃N₄ nanocomposite with a BET surface area of approximately 257 m²/g effectively eliminates Cd ions from aqueous solutions. The faulty sites in Ru-ZnO, which introduce a strong contact between the

defects and the Cd ions, are principally responsible for this extraordinary capacity. Consequently, the defective Ru-ZnO-g-C$_3$N$_4$ nanocomposite displayed a high Cd ion capacity of 475.5 mg g$^{-1}$. Cd ion adsorption capabilities of certain nanocomposites are superior to those of previously researched materials. Significantly, the nanomaterial's effectiveness in removing Cd ions was highlighted by its ability to work in an extensive pH at 5. The kinetic studies of the adsorption process based on the investigated nanocomposite provide corroborating evidence for the chemisorption character of the Cd ions. Ru-ZnO-g-C$_3$N$_4$ nanocomposite flourishes as a feasible adsorbent for removing pollutants in water treatment due to its efficiency and practicability.

**Author Contributions:** Conceptualization, A.M.; M.A.B.A. and S.M.S.; methodology, M.I.; software, A.A.; validation, A.M.; M.A.B.A. and S.M.S.; formal analysis, M.I.; investigation, A.A.; resources, M.I.; data curation, A.A.; writing—original draft preparation, A.M.; writing—review and editing, M.I.; M.A.B.A.; A.M.; and S.M.S.; visualization, A.M.; supervision, A.M.; M.A.B.A. and S.M.S.; project administration, M.I.; funding acquisition, M.I. All authors have read and agreed to the published version of the manuscript.

**Funding:** The authors extend their appreciation to the Deputyship for Research & Innovation, Ministry of Education, Saudi Arabia for funding this research work through the project number (QU-IF-4-5-1-31841).

**Data Availability Statement:** All data and information recorded or analyzed throughout this study are included in this paper.

**Acknowledgments:** The authors extend their appreciation to the Deputyship for Research & Innovation, Ministry of Education, Saudi Arabia for funding this research work through the project number (QU-IF-4-5-1-31841). The authors also thank to Qassim University for technical support.

**Conflicts of Interest:** The authors declare no conflict of interest.

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
