# Peer review of "High Poisonous Cd Ions Removal by Ru-ZnO-g-C3N4 Nanocomposite: Description and Adsorption Mechanism"

_inorganics, doi:10.3390/inorganics11040176_

Round 1

Reviewer 1 Report

The paper presents interesting results on the sorption kinetics of cadmium ions. In water purification processes, when using sorbents, the main characteristics are the effectiveness of sorbents (sorption capacity, reusability) and cost. In the course of reading the text of the manuscript, the reviewer had several comments.

1. Why was this particular sorbent chosen? There is no justification for the choice of this compound in the text.

2. Page 3, subsection 3.1: Based on what x-ray data do the authors conclude that Ru-ZnO is deposited on g-C3N4? How did the authors determine the decrease in q-space? For such a statement, X-ray data for g-C3N4 are needed.

3. Page 4, Fig. 2.: What does the peak at 731 cm-1 refer to? When considering the IR spectra of metal oxides, peaks and bands in the range of 800-400 cm-1 refer to certain vibrations of the Me-O, Me=O, Me-O-Me or O-Me-O bonds.

4. Page 4, line 136: What means size? Is it pore diameter or size of particles? Incorrect use of terminology by the authors. There is no term pore size, there is the concept of pore diameter, since pores can have different shapes. Which pores correspond to the H1 type of hysteresis? What means cc/g?

5. age 5, line 149: TEM image presented in Fig. 4 !

6. Page 9, line 227: "Due to the high number of active sites on the Ru-ZnO-g-C3N4 nanocomposite surface..." What assessments were made for such an assertion?

7. IR spectra data in Fig. 10a are similar to the data in Fig. 2. Page 11, lines 280-282:  "As depicted in Fig. 10a..." How can you see it? The transmission curve after the sorption of cadmium ions in the presented form is uninformative. The fact that cadmium ions were eliminated in the triazine ring should be confirmed by the shift of characteristic peaks.

8. The authors argue that the exceptionally efficient removal of ions is due to the high specific surface area of the sorbent. Please provide specific surface data for other sorbents presented in Table 3.

9. In the title, the authors state the mechanism of adsorption of CD2+ ions by the sorbent. Based on the analysis of metal ion sorption isotherms and sorption kinetics, what is the mechanism? Is it physical sorption or is there a chemical nature?

10. The conclusions do not correspond to the data presented in the text. The conclusions mention recycled up to 5 times but there is no data about this in the text of the manuscript.

11. What are the advantages of the investigated sorbent in comparison with others, excluding the theoretical high sorption capacity?

The manuscript is interesting, but improvements are needed in accordance with the comments in order to increase the readability of the paper.

Author Response

The paper presents interesting results on the sorption kinetics of cadmium ions. In water

purification processes, when using sorbents, the main characteristics are the effectiveness of sorbents (sorption capacity, reusability) and cost.

In the course of reading the text of the manuscript, the reviewer had several comments.

  1. Why was this particular sorbent chosen? There is no justification for the choice of this compound in the text.

In response to this comment the following part was added in 3.3. Assessment study

The nanostructures of the synthesized Ru-ZnO-g-C3N4 can be used as a suitable adsorbent for aqueous cadmium (II) ions due to their simplicity of manufacture, high adsorption efficacy, recovery ability, and reusability.

  1. 2. Page 3, subsection 3.1: Based on what x-ray data do the authors conclude that Ru-ZnO is deposited on g-C3N4? How did the authors determine the decrease in q-space? For such a statement, X-ray data for g-C3N4 are needed.

The X-ray data for g-C3N4 was added.

  1. 3. Page 4, Fig. 2.: What does the peak at 731 cm1 refer to? When considering the IR spectra of metal oxides, peaks and bands in the range of 800-400 cm-1 refer to certain vibrations of the MeO, Me=O, Me-O-Me or O-Me-O bonds.

The attribution of the band at 731 cm-1 was added in the manuscript.

  1. Page 4, line 136: What means size? Is it pore diameter or size of particles? Incorrect use of terminology by the authors. There is no term pore size, there is the concept of pore diameter, since pores can have different shapes. Which pores correspond to the H1 type of hysteresis? What means cc/g?
  2. 5. Page 5, line 149: TEM image presented in Fig. 4 !

corrected

  1. 6. Page 9, line 227: "Due to the high number of active sites on the Ru-ZnO-g-C3N4

nanocomposite surface..." What assessments were made for such an assertion?

Based on the FTIR study, on the Ru-ZnO-g-C3N4 nanocomposite surface possess a high number of active sites on the surface such as -NH2, OH, and s-triazine groups.

  1. 7. IR spectra data in Fig. 10a are similar to the data in Fig. 2. Page 11, lines 280-282: "As depicted in Fig. 10a..." How can you see it? The transmission curve after the sorption of cadmium ions in the presented form is uninformative. The fact that cadmium ions were eliminated in the triazine ring should be confirmed by the shift of characteristic peaks.

FTIR spectra of Ru-ZnO-g-C3N4 before and after the adsorption of Cd ions have been improved. As the spectra show, the fact that the cadmium ions have been eliminated in the triazine ring is confirmed by the shift of the peak at 889 cm-1 (triazine ring mode).

  1. 8. The authors argue that the exceptionally efficient removal of ions is due to the high specific surface area of the sorbent. Please provide specific surface data for other sorbents presented in Table 3.

provided

  1. 9. In the title, the authors state the mechanism of adsorption of CD2+ ions by the sorbent. Based on the analysis of metal ion sorption isotherms and sorption kinetics, what is the mechanism? Is it physical sorption or is there a chemical nature?

In response to this insightful comment the following part was inserted in the manuscript

In line with the pseudo-second-order model, the rate-determining stage is regarded as chemical adsorption that involves the adsorbent/adsorbate of electrons between the adsorbent and adsorbate [47]. The high regression coefficient (R2 = 0.9958) of the Elovich model [48] provides corroborating evidence for the chemisorption character of the Cd ions' adsorption by Ru-ZnO-g-C3N4.

  1. 10. The conclusions do not correspond to the data presented in the text. The conclusions mention recycled up to 5 times but there is no data about this in the text of the manuscript.

In response to this insightful comment the conclusion part was modified in the manuscript

  1. 11. What are the advantages of the investigated sorbent in comparison with others, excluding the theoretical high sorption capacity?

 In response to this comment the following part was added in 3.3. Assessment study

The nanostructures of the synthesized Ru-ZnO-g-C3N4 can be used as a suitable adsorbent for aqueous cadmium (II) ions due to their simplicity of manufacture, high adsorption efficacy, recovery ability, and reusability.

Reviewer 2 Report

In general, the concept of the article is interesting. However, some issues need to be improved before considering the article for publication.

In introduction: “Emissions of inorganic pollutants into the environment are a major cause for concern due primarily to their transformation into more harmful compounds [1].: - please provide the examples of inorganic pollutants and into what more harmful compounds they have transformed.

In Experimental methods:

-please provide suppliers of used reagents

In Figure 6 and 7 - Whether the captions of the figures should not begin with capital letters.

Author Response

In general, the concept of the article is interesting. However, some issues need to be improved before considering the article for publication.

In introduction: “Emissions of inorganic pollutants into the environment are a major cause for concern due primarily to their transformation into more harmful compounds [1].: - please provide the examples of inorganic pollutants and into what more harmful compounds they have transformed.

The following part was inserted in the introduction part:

Cadmium is a metal with toxic characteristics; long-term, superficial intake of cadmium has adverse impacts on the health of people, including the development of diabetes, hypertension, and cancer [2]. Diet is the primary source of cadmium exposure, second only to smoking; multiple studies have linked prolonged dietary consumption of cadmium with a higher risk of renal dysfunction and osteoporosis [3]. Dietary exposure to cadmium occurs when crops that are consumed as food absorb cadmium from agricultural soil. The high persistence of cadmium in soil and the high soil-to-plant transfer rates facilitate this process [4].

In Experimental methods:

-please provide suppliers of used reagents

The authors thank the Reviewer for adding more details in the experimental part. All required details have been now incorporated in the experimental section.

In Figure 6 and 7 - Whether the captions of the figures should not begin with capital letters.

Corrected

Round 2

Reviewer 1 Report

The authors took into account all the comments of the reviewer, increasing the readability of the text. Thank you.